# Supplementation of chicory root powder as an alternative to antibiotic growth promoter on gut pH, gut microflora and gut histomorphometery of male broilers

**Srinivas Gurram** [1]*, **Chinni Preetam. V.**[1], **Vijaya Lakshmi. K.**[2], **Raju. M. V. L. N.** [3], **Venkateshwarlu. M.**[4], **Swathi Bora**[5]

1 Department of Poultry Science, College of Veterinary Science, PV Narsimha Rao Telangana Veterinary University, Hyderabad, India, 2 Department of Livestock Farm Complex, College of Veterinary Science, PV Narsimha Rao Telangana Veterinary University, Warangal, India, 3 Poultry Nutrition, ICAR, Directorate of Poultry Research, Rajendranagar, Hyderabad, India, 4 Department of Animal Nutrition, PV Narsimha Rao Telangana Veterinary University, Hyderabad, India, 5 Department of Veterinary Pathology, College of Veterinary Science, PV Narsimha Rao Telangana Veterinary University, Hyderabad, India

☯ These authors contributed equally to this work.
* gurramsrinivas4@gmail.com

**Data Availability Statement:** All relevant data are within the paper and its Supporting Information files.

## Abstract

The experiment was conducted to study the effect of chicory root powder on the gut performance of broilers. For this purpose, two hundred commercial male broiler chicks were randomly divided into 5 treatment groups with 8 replications of 5 birds each and reared in battery brooders up to 42 days of age. The experimental design consisted of; T1 basal diet (BD) without antibiotic, T2: BD + antibiotic (BMD at 500 gm/ton), T3: BD + chicory root powder (0.5%), T4: BD + chicory root powder (1.0%), T5: BD + chicory root powder (1.5%). The results revealed that supplementation of 1.0% chicory root powder recorded significantly (P<0.05) higher body weight gain, feed intake and better feed conversion ratio (FCR) compared to antibiotic, control and 0.5 & 1.5% chicory powder at 42 days of age. Supplementation of various levels of chicory root powder significantly (P<0.05) lowered (P<0.05) the pH in duodenum, jejunum, ileum and caecum compared to control. Supplementation of chicory root powder (0.5, 1.0 & 1.5%) significantly (P<0.05) decreased the *E. coli* and *Salmonella* counts and (P<0.05) increased the *Lactobacilli* counts in ileum when compared to control and antibiotic groups. Supplementation of chicory (1.0% and 1.5%) groups significantly (P<0.05) increased the villus height (VH), crypt depth (CD), VH:CD ratio and villus width (VW) in the duodenum, ileum and jejunum at 42 d of age. Supplementation of chicory 1.0% and 1.5% groups significantly (P<0.05) increased the goblet cell number in duodenum, jejunum and ileum of broilers. Therefore, chicory root powder (1.0 and 1.5%) can be used as an alternative to antibiotic for improving gut performance of broiler chicken.

**Funding:** The author(s) received no financial support for the research, authorship, and/or publication of this article.

**Competing interests:** The authors have declared that no competing interests exist.

## Introduction

The mode of action of antibiotics is by suppressing the negative effects of pathogenic bacteria in the gastrointestinal tract and increasing nutrients absorption in intestines. But in recent years, due to negative human health issue of antibiotic resistance, there is an increasing pressure to reduce or eliminate the use of antibiotics as growth promoters [1]. The increased awareness among consumers for the poultry products without antibiotic residue encouraged the utilization of suitable alternatives for antibacterial compounds [2]. Recently, products like chicory root powder and coriander seed powder gaining attention as herbal feed additives having antimicrobial properties (Due to inulin). Chicory is a perennial herbaceous plant and their roots are baked, roasted, ground and used as coffee additive. Fresh chicory root contains 11–15% inulin and it may increase up to 40% in dried chicory root powder. Dried chicory root powder is a good source of inulin type fructans and oligofructose chains known for having prebiotic action without any toxicity [3]. Inulin-type fructans are indigestible carbohydrates, recognized as dietary fibers that improve intestinal health and bird's performance through their prebiotic properties [4]. The fermentation activity of inulin inhibits the growth of harmful strains, selectively stimulates the growth of beneficial bacteria by decreasing the intestinal pH through increasing the absorption of short chain fatty acids and thus promotes the growth of broiler chickens [5, 6].

It has also been suggested that feeding chicory root powder to broiler can increase the absorption of nutrients by increasing jejunum villus height and crypt depth [7]. Similarly, probiotic + prebiotic supplementation decreased intestinal pH and viscosity in broilers [8]. Supplementation of prebiotics Increased the intestinal characteristics like villus total and villus height in broilers [9]. Addition of postbiotics and inulin combinations in broilers significantly ($P<0.05$) increased the villus height and crypt depth of the duodenum, ileum and jejunum [10]. In view of the above, this experiment was designed to evaluate the dietary supplementation chicory root powder as an alternative to antibiotic growth promoter on gut pH, gut ecology and morphometry of broilers.

## Materials and methods

For this purpose, 200 day-old male broiler chicks (Vencobb) were distributed randomly in to five dietary treatments of eight replicates with five chicks in each replicate. At day one, chicks were wing banded and reared under optimum brooding conditions. The broilers were maintained in battery brooders with feed and water fed *ad lib* from 1 to 42 days age. The birds were fed with maize and soybean meal-based diets containing 2958, 3074 and 3163 kcal ME and 22.76, 21.58 and 19.68 percent crude protein, respectively during prestarter (0-14d), starter (15-28d) and finisher (28-42d) phases (Table 1). The experimental design consisted of; T1: Basal diet (BD) without antibiotic, T2: BD + antibiotic (Bacitracin Methylene Disalicylate at 500 gm/ton–manufacturer Zoetis), T3: BD + Chicory root powder (0.5%), T4: BD + Chicory root powder (1.0%), T5: BD + Chicory root powder (1.5%). The chemical composition of Chicory root powder was given in Table 2.

One bird from each replicate was sacrificed on 42nd day of age from each treatment group. Gut (proventriculus, gizzard, duodenum and ileum) pH was recorded immediately after collection of gastro-intestinal contents from respective part of gut. Approximately 1.0 g of sample content was suspended in 5ml distilled water, mixed vigorously with glass rod and pH was determined using digital pH meter. The electrode was rinsed with distilled water and recalibrated in between the readings [11].

Record of temperature was maintained on daily basis where the highest daily average temperature recorded is 38.05˚C and the lowest temperature is 18.5˚ C during the experimental

**Table 1. Ingredient composition of basal diets (in kgs) fed to the commercial broilers from 0-42days.**

| Ingredient | Pre-starter (0-14d) | Starter (15-28d) | Finisher (29-42d) |
|---|---|---|---|
| Maize | 56.09 | 56.4 | 59.8 |
| Oil | 2.04 | 4.0 | 5.0 |
| Soyabean meal (CP 46%) | 37.1 | 34.6 | 30.1 |
| Stone grit | 1.58 | 1.83 | 1.88 |
| Dicalcium phosphate | 1.85 | 1.90 | 1.96 |
| Salt (NaCl) | 0.46 | 0.46 | 0.48 |
| DL-Methionine | 0.22 | 0.18 | 0.16 |
| L-Lysine HCl (99%) | 0.17 | 0.15 | 0.13 |
| Trace Mineral Mixture* | 0.10 | 0.10 | 0.10 |
| Vitamin AB2D3K** | 0.020 | 0.020 | 0.020 |
| Vitamin B-Complex** | 0.025 | 0.025 | 0.025 |
| Coccidiostat (Coxynil) | 0.05 | 0.05 | 0.00 |
| Choline chloride (50%) | 0.15 | 0.15 | 0.15 |
| Toxin binder | 0.10 | 0.10 | 0.10 |
| **Total** | **100** | **100** | **100** |
| Nutrient composition (calculated values) | | | |
| ME (kcal/kg) | 2958 | 3074 | 3163 |
| Crude protein (%) | 22.76 | 21.58 | 19.68 |
| Lysine (%) | 1.30 | 1.21 | 1.08 |
| Methionine (%) | 0.55 | 0.49 | 0.45 |
| Calcium (%) | 0.97 | 1.04 | 1.06 |
| Available phosphorous (%) | 0.45 | 0.45 | 0.45 |

*Trace mineral provided per kg diet: Manganese 120mg, Zinc 80mg, Iron 25mg, Copper 10mg, Iodine 1mg and Selenium 0.1mg.

**Vitamin premix provided per kg diet: Vitamin A 200000IU, Vitamin D3 12000IU, Vitamin E 10mg, Vitamin K 2mg, Riboflavin 25mg, Vitamin B1 1mg, Vitamin B6 2mg, Vitamin B12 40mg and Niacin 15mg.

period. The average relative humidity is 65.56 during the experimental period. The experiment was conducted during December and January—2020.

## Gut ecology

Eight birds from each dietary treatment were slaughtered on 42nd day and intestines were dissected at Meckel's diverticulum. Approximately 5g of ileal digesta was collected aseptically into sterile sampling tubes and immediately transferred on ice to the laboratory for microbiological examination for *E. coli*, *Salmonella* spp and *Lactobacilli* spp counts. Eosin methylene blue agar

**Table 2. Chemical composition of chicory root powder.**

| Composition (%) | Chicory root powder |
|---|---|
| Moisture | 3.16% |
| Crude protein | 14.55 |
| Fat | 1.76 |
| ash | 3.98 |
| Crude fiber | 30.01 |
| Total carbohydrates | 48.76 |
| Inulin | 46.89 |

(EMB) was used for *E. coli* growth, Salmonella-Shigella agar (SS Agar) used for *Salmonella* spp. and MRS agar (De Man, Rogosa and Sharpe agar) used for *Lactobacilli* spp growth.

Then, 9 sterile test tubes with lids containing 9mL of phosphate buffer solution (PBS, pH-7.4) as diluent were prepared. Approximately 1g of the intestinal contents taken by sterile swab and homogenized for 3 min, aseptically mixed, added to the tubes, and diluted up to $10^9$. Later, 1ml of the contents of each test tube was transferred to one of three selective agar media on petri plates, respectively [12]. Aerobic bacterial plates (*E. coli*, *Salmonella* spp) were placed in an incubator at 37˚C for 24 hours. Anaerobic (*Lactobacilli* spp) medium plates were placed in an anaerobic jar with an anaerobic gas pack system at 37˚C for 24 hours. Finally, the intestinal bacterial colony populations formed in each plate was counted by colony counter and the number of colonies was expressed as log10 value.

## Histomorphometry

On 42$^{nd}$ day during slaughter, 2 cm long segment of duodenum, jejunum and ileum of six birds from each treatment were collected and then washed with physiological saline solution and fixed in 10% neutral buffered formalin solution. These samples were processed for histo-morphological examination in terms of measurement of parameters like villous height (VH), cryptal depth (CD), villus width and villous height:crypt depth ratio. Histological technique involves processes like fixation of tissue, dehydration, clearing, embedding, cutting and staining. Fixation in 10% formalin with approximately 10–20 times the volume of the specimen was done. Tissues were dehydrated by using increasing strength of alcohol like 50%, 70%, 90% and 100%. Clearing was done by replacing alcohol by xylene for 0.5–1 hour. Impregnation of tissue with wax was done at melting point temperature of paraffin wax and the volume of wax was about 25–30 times the volume of tissues for a total duration of 4 hours. Impregnated tissues were placed in a mould with their labels and then fresh melted wax was poured in it and allowed to settle and solidify. These paraffin embedded tissues were sectioned at 5μm thickness and stained routinely with Hematoxylin-Eosin stain (H&E).

Histological sections were examined under 2X of light microscopy with micrometry and photographic attachment. The images were analyzed using image analyzing software (OLYM-PUS cell Sens Standard, version 1.13). A total of 20 intact well oriented crypt-villous units per bird were selected randomly, measured and the mean length was calculated for each sample. Villous height was measured from the tip of the villi to the base between individual villi, and crypt depth measurements were taken from the valley between individual villi to the basal membrane.

Data analyzed for mean, standard errors and analysis of variance as per method of [13] and comparison of means were done [14] using software of Statistical Package for Social Sciences (SPSS) 20.0 version and significance was considered at P<0.05.

## Ethical approval

All authors hereby declare that all biological trials have been examined and approved by the ethics committee of PV Narsimha Rao Telangana Veterinary University, Rajendranagar, Hyderabad, India (Institutional Animal Ethics Committee number: IV/2019-02/IAEC/CVSC, Hyderabad, India) and have therefore been performed in accordance with the ethical standards.

## Results and discussion

Groups supplemented with CRP (1.0%) had significantly (P<0.05) higher weight gain compared to antibiotic, control and CRP (0.5 & 1.5%) treatment groups at 42 days of age (Table 3).

**Table 3. Effect of chicory root powder on body weight gain (g), feed intake and feed conversion ratio of broiler chicken.**

| Trt | Diets | Body weight gain | Feed intake | Feed conversion ratio |
|-----|-------|------------------|-------------|------------------------|
| T[1] | Control | 2140[c] | 3654[bc] | 1.708[c] |
| T[2] | Antibiotic | 2214[b] | 3677[abc] | 1.663[a] |
| T[3] | CRP (0.5%) | 2202[b] | 3711[ab] | 1.684[b] |
| T[4] | CRP (1.0%) | 2263[a] | 3729[a] | 1.648[a] |
| T[5] | CRP (1.5%) | 2169[c] | 3648[c] | 1.683[b] |
| | SEM | 8.278 | 9.595 | 0.0042 |
| | N | 8 | 8 | 8 |
| | *p*-value | **0.001** | **0.020** | **0.001** |

Supplementation of 0.5% and 1.0% CRP significantly (P<0.05) increased the feed intake compared to antibiotic, control and 1.5% CRP groups. The best feed efficiency was recorded with CRP (1.0%) and antibiotic groups followed by CRP (0.5%) and CRP (1.5%), where as poor feed efficiency was noticed in control group at 42 d of age. The significant (P<0.05) improvement in body weight gain and feed efficiency by feeding the chicory root powder (1.0%) is in accordance with the earlier findings of Yusrizal and Chen (2003a) [15]; Karwan *et al.* (2016) [10]; Yousfi *et al.* (2017) [16] and Praveen *et al.* (2017) [17]. Supplementation of chicory root powder (1% and 3% levels) significantly (P<0.05) improved overall body weight gain than control [7]. Nabizadeh (2012) [18] stated that addition of inulin @ 1.0% level significantly (P<0.05) increased live body weight and feed efficiency of broilers at 42 d of age. However, contrary to our findings, Liu *et al.* (2011) [19] observed no significant (P>0.05) difference with supplementation of chicory root powder in broiler diets. It was hypothesized that the beneficial effect of chicory root powder in broilers might be due to prebiotic properties of inulin. Inulin-type fructan is a soluble fermentable fiber that is not digested by host digestive enzymes and serves as a substrate for beneficial bacteria in the gut of birds [20]. The fermentation activity inhibits the growth of *Escherichia coli* and *Salmonella* and selectively stimulates the growth of *Bifidobacteria* and *Lactobacillus* development in the gut [15].

## Gut pH

pH values of various segments of the digestive system (except proventriculus) were significantly (P<0.05) influenced at the time of slaughter age (42 d). Supplementation of various levels of chicory root powder (0.5, 1.0 &1.5%) significantly (P<0.05) lowered the pH in duodenum, ileum, and caecum when compared to control and antibiotic groups (Table 4).

**Table 4. Effect of chicory root powder on gut pH of broiler chicken at 42 d of age.**

| Trt | Diets | Proventriculus | Duodenum | Jejunum | Ileum | Caecum |
|-----|-------|----------------|----------|---------|-------|--------|
| T[1] | Control | 3.71 | 5.94[c] | 6.63[c] | 6.86[c] | 7.30[b] |
| T[2] | BMD | 3.68 | 5.71[b] | 6.44[b] | 6.52[b] | 7.21[b] |
| T3 | CRP (0.5%) | 3.68 | 5.41[a] | 6.39[b] | 6.40[a] | 6.90[a] |
| T4 | CRP (1.0%) | 3.66 | 5.39[a] | 6.38[b] | 6.38[a] | 6.83[a] |
| T5 | CRP (1.5%) | 3.70 | 5.45[a] | 6.21[a] | 6.31[a] | 6.79[a] |
| | SEM | 0.031 | 0.038 | 0.025 | 0.034 | 0.038 |
| | N | 8 | 8 | 8 | 8 | 8 |
| | *p*-value | **0.988** | **0.001** | **0.001** | **0.001** | **0.001** |

Value bearing different superscripts within a column are significantly (P<0.05) different

However, supplementation of chicory (1.5%) group significantly (P<0.05) lower jejunum pH values compared to all other treatments. However, the pH values in chicory 0.5% and 1.0% supplemented groups were lower than the control and comparable with antibiotic. In agreement with the results of chicory root powder in this experiment, other researchers [18, 21] reported that supplementation of chicory inulin significantly (P<0.05) decreased the caecum pH in broilers compared to control. In contrast with present study, addition of chicory forage and root inclusion did not effect on caecal pH of broilers at 31 d of age [19]. The decreased intestinal pH might be due to beneficial intestinal microflora, such as *Lactobacillus* spp. or *Bifidobacterium* spp. They use inulin or oligofructose for fermentation to produce short chain fatty acids (acetate, propionate, butyrate and lactate) which creates an acidic environment in GIT of birds [15].

## Gut ecology

Supplementation of antibiotic and chicory treatment groups significantly (P<0.05) decreased the *E. coli* and *Salmonella* counts compared to control at 42 d of age. The lowest *E. coli* counts were recorded in antibiotic group followed by chicory (1.0% & 1.5%) and chicory (0.5%) groups (Table 5). In agreement with the lowered *E. coli* and *Salmonella* counts in chicory groups [6], addition of prebiotics and antibiotics (BMD) lowered the total anaerobes and coliforms counts in broilers. Chicory root powder contains a fermentable fiber (Inulin) that is not digested by enzymes and serves as a substrate for beneficial bacteria in the gut. The fermentation activity inhibits the growth of *E. coli* and *Salmonella* and selectively stimulate the growth of *Lactobacillus* and *Bifidobacteria* in the gut [20]. Similarly, Yusrizal and Chen (2003a) [15] indicated that normal intestinal microflora, such as *Lactobacillus* spp. or *Bifidobacterium* spp., use chicory inulin for fermentation more efficiently than other groups of bacteria; these micro-organisms produce short chain fatty acids (SCFA) and lactate on inulin to create an acidic environment which suppresses the growth of *E. coli* and *Salmonella*. The reduction of pathogenic bacteria in the experiment may be due to the increased acidic pH in the intestines. On contrary, Wang *et al.* (2018) [22] reported prebiotics and antibiotic supplementation did not reduce caecal total anaerobic bacteria in broilers. Similarly, Nabizadeh (2012) [18] observed that inulin supplementation did not affect ileal *E. coli* counts in broilers. In agreement with the results of this study, a series of earlier studies demonstrated that addition chicory root powder [10, 21, 23] decreased the harmful bacterial count in broilers.

**Table 5. Effect of chicory root powder on gut microbiota ($log_{10}$ of cfu/g count) in ileum sample of broiler chicken.**

| Trt | Diets | *Escherichia coli* ($log_{10}$ cfu/g) * | *Salmonella* spp ($log_{10}$ cfu/g) ** | *Lactobacillus* spp ($log_{10}$ cfu/g) * |
|---|---|---|---|---|
| $T_1$ | Control | 8.10[d] | 4.53[d] | 7.75[b] |
| $T_2$ | BMD | 6.74[a] | 3.27[a] | 6.77[c] |
| $T_3$ | CRP (0.5%) | 7.07[c] | 4.05[c] | 7.79[b] |
| $T_4$ | CRP (1.0%) | 6.96[b] | 3.91[b] | 7.93[a] |
| $T_5$ | CRP (1.5%) | 6.92[b] | 4.06[c] | 7.89[a] |
| SEM | | 0.078 | 0.067 | 0.070 |
| N | | 8 | 8 | 8 |
| *p*-value | | **0.001** | **0.002** | **0.001** |

Value bearing different superscripts within a column are significantly (P<0.05) different

* Calculated as per $log_{10}$ colony forming units/gram of sample ($10^6$).

** Calculated as per $log_{10}$ colony forming units/gram of sample ($10^3$).

Supplementation of various levels chicory and control group significantly (P<0.05) increased the *Lactobacilli* counts in the ileum compared to antibiotic. The highest *Lactobacilli* counts were recorded in chicory 1.0% level and 1.5% level followed by chicory 0.5% and control groups. In agreement with the increased *Lactobacilli* counts in chicory groups, Yusrizal and Chen (2003a) [15] reported that addition of chicory fructans increased the *Lactobacilli* counts in the gizzard and small intestine contents of broilers. Similarly, Xu *et al.* (2003) [23] observed the enhanced growth of intestinal *Bifidobacterium* and *Lactobacillus* count with fructooligosaccharide supplementation in broilers. Corroborating the results of the present study, several researchers reported that addition of prebiotics increased the *Lactobacillus* counts in the ileum of broilers [5, 6, 22]. This might be due to inulin (fermentable fiber) in chicory root powder that is not digested by enzymes and serves as a substrate for beneficial bacteria in the gut. The fermentation activity inhibits the growth of *E. coli* & *Salmonella* and selectively stimulate the growth of *Lactobacillus* and *Bifidobacteria* in the gut [20]. On contrary, Nabizadeh (2012) [18] reported inulin supplementation had no effect on *Bifidobacteria* and *Lactobacilli* counts in ileum of broilers.

## Gut histomorphometry

Supplementation of chicory (1.0%) level significantly (P<0.05) increased the duodenal villus height (VH), crypt depth (CD), VH:CD ratio and villus width (VW) when compared to control and antibiotic groups (Table 6). The duodenal morphometry parameters in chicory 0.5% and 1.5% groups were intermediate and was statistically comparable with antibiotic group. Significantly (P<0.05) higher jejunal VH and VH:CD ratio was recorded in chicory 1.0% group compared to chicory (0.5 and 1.5%), control and antibiotic groups (Table 7 and Figs 1–5). However, all chicory supplemented groups significantly (P<0.05) increased the jejunal VW compared to control and antibiotic groups. Supplementation of chicory (0.5, 1.0 and 1.5%) diets significantly (P<0.05) increased the ileal villi height (VH), crypt depth (CD) and VH:CD ratio in comparison with control and antibiotic groups (Table 8). Among all the treatments, the highest VH and CD was recorded in chicory (1.0%) group. Supplementation of chicory 1.0% and 1.5% groups significantly (P<0.05) increased the goblet cell number in duodenum, jejunum and ileum of broilers. chicory root powder may reduce the growth of many pathogenic and non-pathogenic intestinal bacteria thereby resulting in reduction in intestinal colonization and infectious process which ultimately decrease the inflammatory process of intestinal mucosa resulting in improved villus height and villus width which in turn improves digestive secretory function and absorption of nutrients [6]. It is hypothesised that the increase

**Table 6. Effect of chicory root powder on histomorphometry of duodenum in broiler chicken.**

| Trt | Diets | Villus height (µm) | Crypt depth (µm) | Villus height: Crypt depth Ratio | Villus width (µm) | Goblet cell number |
|---|---|---|---|---|---|---|
| T$_1$ | Control | 1683.46[b] | 216.99[d] | 7.83[a] | 225.00[c] | 6.50[d] |
| T$_2$ | BMD | 1583.82[c] | 260.37[b] | 6.19[b] | 338.21[b] | 9.17[c] |
| T$_3$ | CRP (0.5%) | 1595.22[c] | 265.26[b] | 6.04[b] | 378.94[a] | 10.33[b] |
| T$_4$ | CRP (1.0%) | 1878.08[a] | 306.77[a] | 6.14[b] | 377.93[a] | 11.83[a] |
| T$_5$ | CRP (1.5%) | 1739.96[b] | 242.54[c] | 7.50[a] | 217.57[c] | 10.67[b] |
| | SEM | 21.767 | 5.606 | 0.160 | 13.624 | 0.374 |
| | N | 6 | 6 | 6 | 6 | 6 |
| | *p*-value | 0.001 | 0.001 | 0.001 | 0.001 | 0.001 |

Value bearing different superscripts within a column are significantly (P<0.05) different

**Table 7. Effect of chicory root powder on histomorphometry of jejunum in broiler chicken.**

| Trt | Diets | Villus height (µm) | Crypt depth (µm) | Villus height: Crypt depth Ratio | Villus width (µm) | Goblet cell number |
|---|---|---|---|---|---|---|
| $T_1$ | Control | 1202.51[d] | 185.81 | 6.47[b] | 183.01[b] | 8.01[c] |
| $T_2$ | BMD | 1303.61[c] | 192.48 | 6.78[b] | 200.54[b] | 10.67[b] |
| $T_3$ | CRP (0.5%) | 1546.11[ab] | 215.62 | 7.19[b] | 258.50[a] | 11.83[b] |
| $T_4$ | CRP (1.0%) | 1626.77[a] | 195.66 | 8.41[a] | 252.78[a] | 14.17[a] |
| $T_5$ | CRP (1.5%) | 1510.07[b] | 208.24 | 7.41[a] | 257.56[a] | 13.67[a] |
| | SEM | 31.674 | 3.890 | 0.181 | 6.488 | 0.458 |
| | N | 6 | 6 | 6 | 6 | 6 |
| | *p*-value | 0.001 | 0.088 | 0.003 | 0.001 | 0.001 |

Value bearing different superscripts within a column are significantly (P<0.05) different

in beneficial microbial activity resulting from dietary chicory root powder supplementation may influence gut morphology and consequently affect gut maturation.

In agreement with the increased gut morphology parameter in chicory supplemented groups, Yusrizal and Chen (2003a) [15] recorded increased jejunum villi distribution with inulin supplementation in broilers. Similarly, Xu *et al.* (2003) [23] reported increased ileal villus height, jejunal and ileal microvillus height and VH:CD ratios at the jejunum and ileum with fructooligosaccharide supplementation in broilers. Izadi *et al.* (2013) [7] observed that feeding chicory root powder (1% and 3%) to broilers significantly (P<0.05) increased the villus length, villus surface area, number of goblet cells and villus length/crypt depth ratio in the jejunum. Supplementation of MOS and FOS in broiler increased the intestinal crypt depth when compared to control and antibiotic groups was reported by Biswas *et al.* (2018) [6]. The positive effect of chicory root powder on the intestinal morphology mainly arose from its ability to create a favourable intestinal environment which had a better effect on intestinal morphology [23]. Prebiotics increase production of fatty acids and reduce intestinal pH. Hence, beneficial effects on intestinal tissue health and morphology are achieved. Corroborating the results of

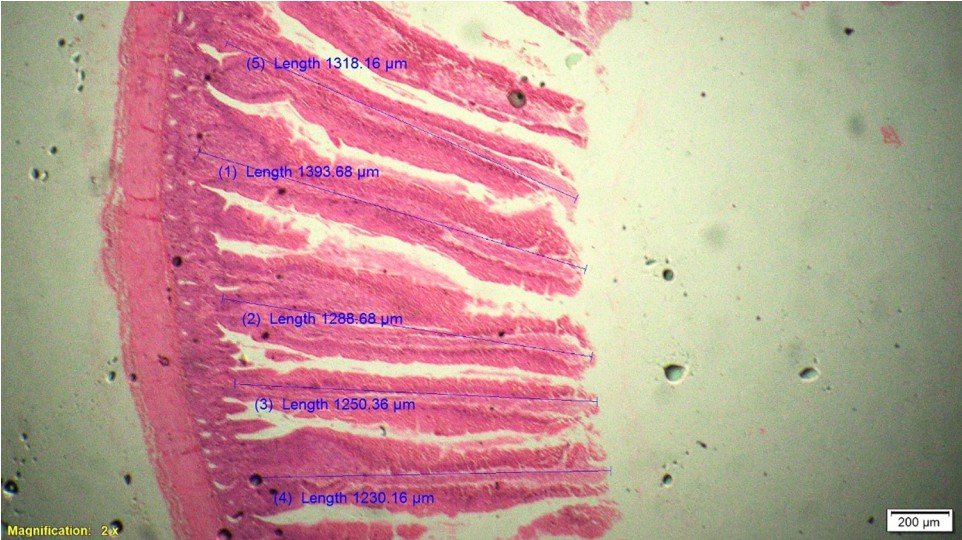

**Fig 1. Photomicrograph of the cross section of jejunum from control group (T1), Hematoxylin-Eosin stain (H&E), 2x.**

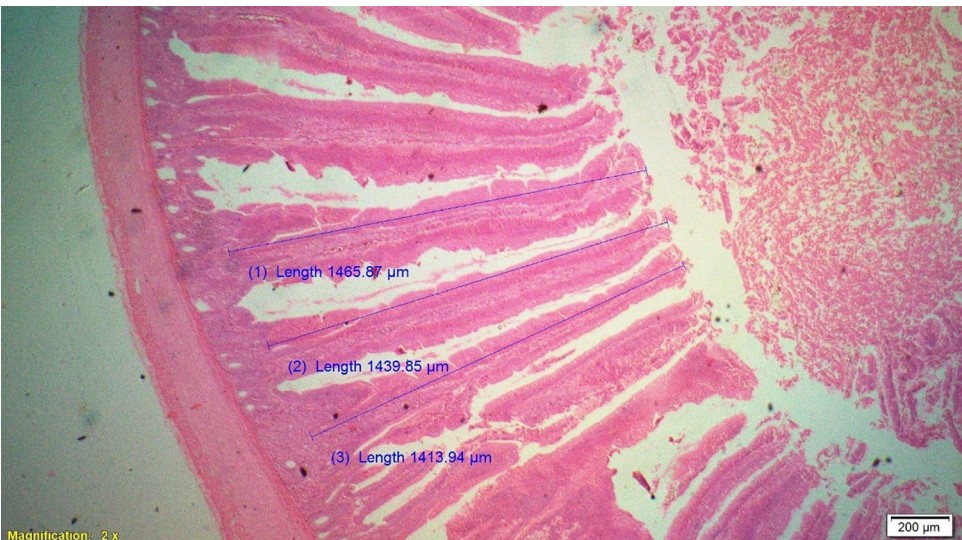

**Fig 2. Photomicrograph of the cross section of jejunum from antibiotic group (T2), H&E, 2x.**

the present study, several researchers reported that addition of prebiotics increased the intestinal morphometry parameters of broilers [6, 9, 10, 18].

Supplementation of different dietary treatments did not affect jejunum CD and ileal VW at 42 d of age. Similarly, Ortiz *et al.* (2009) [24] and Rebole *et al.* (2010) [5] reported inulin supplementation had no effect on VH, CD and VW in the jejunum. Liu *et al.* (2011) [19] observed that chicory forage and root inclusion had no effect on jejunum morphology in broilers.

## Conclusion

Finally, it can be concluded that supplementation of 1.0% chicory root powder was more effective in terms of higher weight gain, better FCR compared to other groups. Dietary

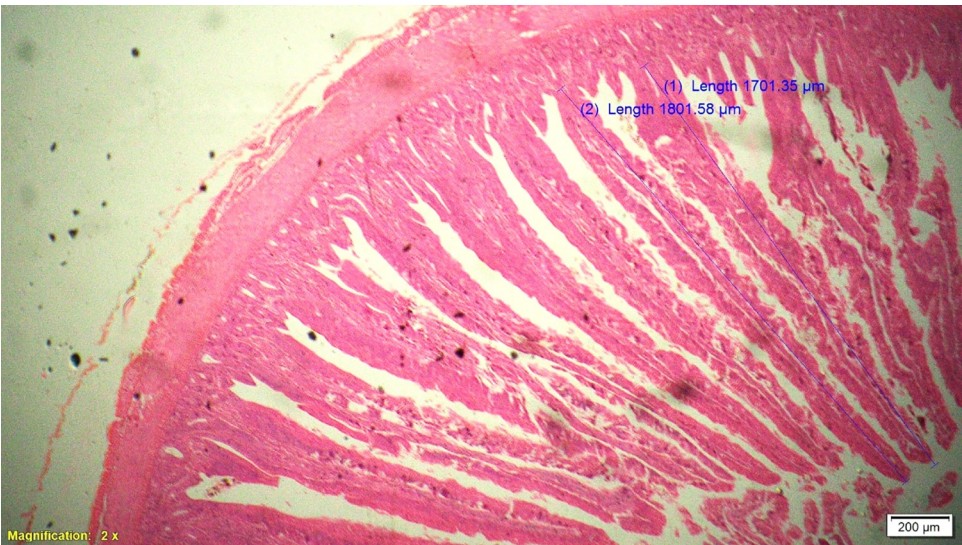

**Fig 3. Photomicrograph of the cross section of jejunum from 0.5% chicory root powder group (T3), H&E, 2x.**

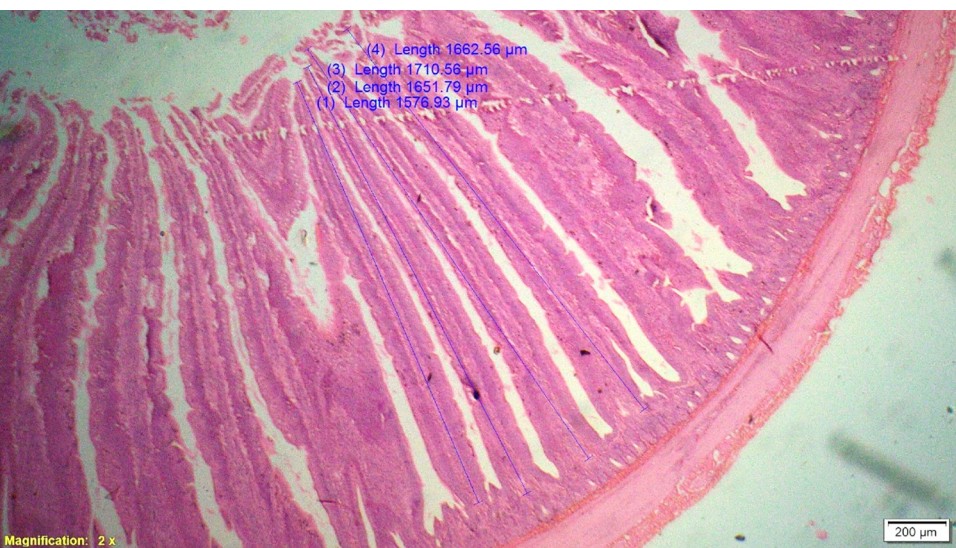

**Fig 4. Photomicrograph of the cross section of jejunum from 1.0% chicory root powder group (T4), H&E, 2x.**

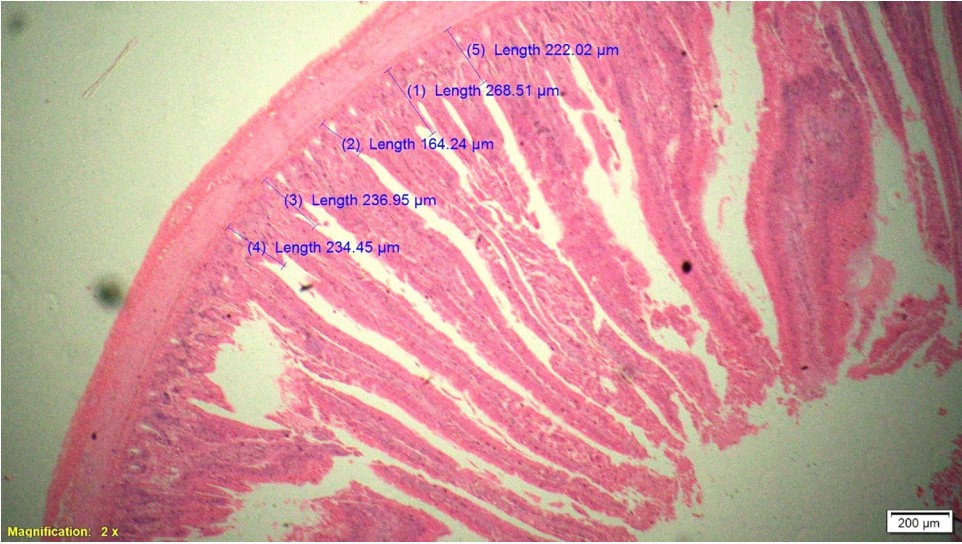

**Fig 5. Photomicrograph of the cross section of jejunum from 1.5% chicory root powder group (T5), H&E, 2x.**

**Table 8. Effect of chicory root powder on histomorphometry of ileum in broiler chicken.**

| Trt | Diets | Villus height (µm) | Crypt depth (µm) | Villus height: Crypt depth Ratio | Villus width (µm) | Goblet cell number |
|---|---|---|---|---|---|---|
| T$_1$ | Control | 863.77[e] | 152.36[c] | 5.82[b] | 175.47 | 10.33[c] |
| T$_2$ | BMD | 1030.35[d] | 207.91[b] | 4.98[d] | 181.18 | 13.83[b] |
| T$_3$ | CRP (0.5%) | 1151.16[b] | 213.67[b] | 5.39[c] | 183.53 | 15.83[a] |
| T$_4$ | CRP (1.0%) | 1579.88[a] | 239.14[a] | 6.64[a] | 180.63 | 17.50[a] |
| T$_5$ | CRP (1.5%) | 1095.46[c] | 161.40[c] | 6.91[a] | 177.28 | 16.83[a] |
| | SEM | 44.631 | 6.526 | 0.142 | 1.810 | 0.533 |
| | N | 6 | 6 | 6 | 6 | 6 |
| | *p*-value | 0.001 | 0.001 | 0.001 | 0.668 | 0.001 |

Value bearing different superscripts within a column are significantly (P<0.05) different

supplementation of chicory root powder at 1.0% and 1.5% level significantly decreased the gut pH and reduced the harmful bacterial count in the intestines of broilers. And also increased the beneficial bacterial count and gut morphometry parameters like VH, CD, VW and goblet cell number in the small intestine. Hence, chicory at 1.0 and 1.5% level can be used as an alternative to antibiotic growth promoter in broiler chicken.

## Supporting information

**S1 Data.**
(XLSX)

## Acknowledgments

The presented manuscript is a part of the first Author's PhD dissertation. The authors are thankful to Department of Poultry Science, College of Veterinary science, PV Narsimha Rao Telangana Veterinary University, R'nagar, Hyderabad, India.

## Author Contributions

**Conceptualization:** Srinivas Gurram.

**Data curation:** Srinivas Gurram.

**Formal analysis:** Srinivas Gurram.

**Investigation:** Chinni Preetam. V., Raju. M. V. L. N., Venkateshwarlu. M.

**Methodology:** Chinni Preetam. V., Venkateshwarlu. M.

**Project administration:** Srinivas Gurram, Raju. M. V. L. N.

**Supervision:** Vijaya Lakshmi. K., Raju. M. V. L. N.

**Validation:** Srinivas Gurram.

**Visualization:** Srinivas Gurram.

**Writing – original draft:** Srinivas Gurram.

**Writing – review & editing:** Srinivas Gurram, Swathi Bora.

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
