## [Decision Letter · Decision Letter 0]

27 Jul 2021

PONE-D-21-14465

SUPPLEMENTATION OF CHICORY ROOT POWDER AS AN ALTERNATIVE TO ANTIBIOTIC GROWTH PROMOTER ON GUT pH, GUT MICROFLORA AND GUT HISTOMORPHOMETERY OF MALE BROILERS

PLOS ONE

Dear Dr. srinivas,

Thank you for submitting your manuscript to PLOS ONE. After careful consideration, we feel that it has merit but does not fully meet PLOS ONE’s publication criteria as it currently stands. Therefore, we invite you to submit a revised version of the manuscript that addresses the points raised during the review process.

It is very important to research on animal sources of alternative plant sources and their derivatives to antibiotics that are banned for use as growth factors in poultry nutrition. However, the flawed experimental design and methodology greatly reduces the merits of this study.

# The introduction section are given the information about the choice of the treatments. Just a small general paragraph (“The increased awareness among consumers for the poultry products without antibiotic residue encouraged the utilization of suitable alternatives for antibacterial compounds”) informs that there are “antibacterial compounds” to be used. In the research, no antibacterial compound was used, but chicory root powder was used. Moreover, what are the biological active substances of this herb root powder? If only a experimental plan could be built on these compounds, the approach would be more accurate.

# According to which criteria the nutritional needs of the animal are determined, the animal's house environment humidity and temperatures, the litter used, etc. such as environmental factors could be given.

#Vitamin and mineral levels in the diet are given incorrectly.

# It is not clear whether the nutritional composition of the basal feed was calculated or analysed. Other feed additives should not be used in feed ingredients. In addition, the coccidiostat used in the diet is of antibiotic origin and cannot be used in finisher diets and is prohibited. Due to the danger of antibiotic residues in the tissues, the anticoccidiostat is removed from the diets of the broilers 7-15 days before slaughter.

# Since this research stands out as a preliminary experimental research, due to methodological deficiencies and flaws in terms of scientific quality.

# I would like to state that the criticisms of the referees came to the fore, especially with the fact that performance data should be given and the number of animals was insufficient.

# Ethics committee document should be provided.

We look forward to receiving your revised manuscript.

Kind regards,

Arda Yildirim, Ph.D.

Academic Editor

PLOS ONE

Additional Editor Comments (if provided):

It is very important to research on animal sources of alternative plant sources and their derivatives to antibiotics that are banned for use as growth factors in poultry nutrition. However, the flawed experimental design and methodology greatly reduces the merits of this study.

# The introduction section are given the information about the choice of the treatments. Just a small general paragraph (“The increased awareness among consumers for the poultry products without antibiotic residue encouraged the utilization of suitable alternatives for antibacterial compounds”) informs that there are “antibacterial compounds” to be used. In the research, no antibacterial compound was used, but chicory root powder was used. Moreover, what are the biological active substances of this herb root powder? If only a experimental plan could be built on these compounds, the approach would be more accurate.

# According to which criteria the nutritional needs of the animal are determined, the animal's house environment humidity and temperatures, the litter used, etc. such as environmental factors could be given.

#Vitamin and mineral levels in the diet are given incorrectly.

# It is not clear whether the nutritional composition of the basal feed was calculated or analysed. Other feed additives should not be used in feed ingredients. In addition, the coccidiostat used in the diet is of antibiotic origin and cannot be used in finisher diets and is prohibited. Due to the danger of antibiotic residues in the tissues, the anticoccidiostat is removed from the diets of the broilers 7-15 days before slaughter.

# Since this research stands out as a preliminary experimental research, due to methodological deficiencies and flaws in terms of scientific quality.

# I would like to state that the criticisms of the referees came to the fore, especially with the fact that performance data should be given and the number of animals was insufficient.

# Ethics committee document should be provided.

Thanks.

Reviewers' comments:

Reviewer's Responses to Questions

**Comments to the Author**

1. Is the manuscript technically sound, and do the data support the conclusions?

Reviewer #1: Yes

Reviewer #2: Yes

Reviewer #3: Partly

2. Has the statistical analysis been performed appropriately and rigorously? 

Reviewer #1: Yes

Reviewer #2: Yes

Reviewer #3: N/A

3. Have the authors made all data underlying the findings in their manuscript fully available?

Reviewer #1: Yes

Reviewer #2: Yes

Reviewer #3: Yes

4. Is the manuscript presented in an intelligible fashion and written in standard English?

Reviewer #1: Yes

Reviewer #2: Yes

Reviewer #3: Yes

5. Review Comments to the Author

Reviewer #1: in material and methods which requirements used to adjust feeding programme. in all feeding programm starter, grower and finisher if use prestarter used for 1st week and protein content 24%. so i see you can use starter(1-14) or follow the recommendation book of the chickens. in table of ingredients which type of Soya bean used(44-46-48). vitamin premix try to check its composition vitamin A may be 10000000 IU.

Nutrient composition calculated or analyzed?

Chicory root powder composition and its content of inulin and other essential oils or contents if possible.

BMD composition and concentration and manufacture.

digital pH meter which type its model?

in table 2 or table 3 use the same groups name Chicory (0.5%) or CRP (0.5%) and not use antibiotic may use BMD.

if possible insert dates about growth performance table and the effects of these Chicory root powder in feed intake and gain and FCR and other parameters to show effect of these treatments on broiler performance

Reviewer #2: Dear Authors Regarding the manuscript title SUPPLEMENTATION OF CHICORY ROOT POWDER AS AN ALTERNATIVE TO ANTIBIOTIC GROWTH PROMOTER ON GUT pH, GUT MICROFLORA AND GUT HISTOMORPHOMETERY OF MALE BROILERS

The scientific background of the topic was well mentioned in the introduction part. The experiment design, as well as the replicates and methods used, were very good. The results obtained were presented in tables well discussed with other author’s results. However, some observation in the present paper should be corrected and add to improve the quality of the paper.

• Table 1 Ingredients and nutrient composition of the experimental diets, need to carful checking for the following :

1- SBM , add the crude protein level 44 or 46%?

2- MEn, kcal/kg, Ca, digP , not according to the Vencobb requirement.

• Introduction and Discussion

Need some other references about using antibiotic and its alternatives in broilers. I recommend you read the following references:

Ahmed A. Saleh, K. Amber and A. A. Mohammed (2020) Dietary supplementation with avilamycin and Lactobacillus acidophilus effects growth performance and the expression of growth-related genes in broilers. Animal Production Science. 60(14) 1704-1710

Ahmed A. Saleh , Tarek A. Ebeid , Alaeldein M. Abudabos. (2018) Effect of dietary phytogenics (herbal mixture) supplementation on growth performance, nutrient utilization, antioxidative properties and immune response in broilers. Environmental Science and Pollution Research. 25:14606–14613.

Ahmed. A. Saleh, Daichi Ijiri and Akira Ohtsuka (2014). Effects of summer shield supplementation on the growth performance, nutrient utilization, and plasma lipid profiles in broiler chickens. Journal of Veterinarni Medicina. 59, (11): 536–542.

• Results: it will be better if you present the data for performance in table.

Reviewer #3: Material and methods:

The author stated that in part of material and methods )One bird from each replicate was sacrificed on 42nd day of age from each treatment group. Gut (proventriculus, gizzard, duodenum and ileum) pH was recorded immediately after collection of gastro-intestinal contents from respective part of gut( .

The number of samples of one sample is very, very, very small. We cannot statistically analyze and rely on these results to generalize their results and be confident in their results. The number should not be less than three samples as a minimum.

6. PLOS authors have the option to publish the peer review history of their article (what does this mean?). If published, this will include your full peer review and any attached files.

Reviewer #1: **Yes: **Hamada Ahmed

Reviewer #2: No

Reviewer #3: **Yes: **Lamiaa M. Radwan

---

## [Author Response · Author response to Decision Letter 0]

29 Sep 2021

the manuscript has been modified as per reviewers and editor suggestion (enclosed file)

SUPPLEMENTATION OF CHICORY ROOT POWDER AS AN ALTERNATIVE TO ANTIBIOTIC GROWTH PROMOTER ON GUT pH, GUT MICROFLORA AND GUT HISTOMORPHOMETERY OF MALE BROILERS

Reviewer comments Reply by author

The introduction section are given the information about the choice of the treatments. Just a small general paragraph (“The increased awareness among consumers for the poultry products without antibiotic residue encouraged the utilization of suitable alternatives for antibacterial compounds”) informs that there are “antibacterial compounds” to be used. In the research, no antibacterial compound was used, but chicory root powder was used. Moreover, what are the biological active substances of this herb root powder? If only a experimental plan could be built on these compounds, the approach would be more accurate. • The anti-microbial activity of chicory root powder due to inulin which contains prebiotic action

• Introduction was modified as per reviewer suggestion

• Biological active substance in chicory powder is - Inulin

According to which criteria the nutritional needs of the animal are determined, the animal's house environment humidity and temperatures, the litter used, etc. such as environmental factors could be given. • Temperature and humidity has been included in results.

• The experiment was conducted in battery brooders. 

Vitamin and mineral levels in the diet are given incorrectly. Vitamin and mineral levels were added as per vencobb requiements.

It is not clear whether the nutritional composition of the basal feed was calculated or analysed. Other feed additives should not be used in feed ingredients. 

In addition, the coccidiostat used in the diet is of antibiotic origin and cannot be used in finisher diets and is prohibited. Due to the danger of antibiotic residues in the tissues, the anticoccidiostat is removed from the diets of the broilers 7-15 days before slaughter. • nutritional composition of the basal feed was calculated values

• sir, we have used herbal origin coccidiostat that is coxynil

• moreover, we have withdrawn coccidiostats from finisher diets

Since this research stands out as a preliminary experimental research, due to methodological deficiencies and flaws in terms of scientific quality. • Chicory contains about 40 % inulin and inulin type fructans and oligofructose chains known for having prebiotic action and exert antimicrobial action. Based on this, the experiment was designed

I would like to state that the criticisms of the referees came to the fore, especially with the fact that performance data should be given and the number of animals was insufficient. • Sir, performance data is going to publish in other journal. That’s why it was not included in journal

Ethics committee document should be provided. • Sir, will be submitted during resubmission time

Reviewer #1:

Reviewer comments Reply by author

in material and methods which requirements used to adjust feeding programme. in all feeding programm starter, grower and finisher if use prestarter used for 1st week and protein content 24%. so i see you can use starter(1-14) or follow the recommendation book of the chickens.

 in table of ingredients which type of Soya bean used(44-46-48). vitamin premix try to check its composition vitamin A may be 10000000 IU.

 The birds were fed with maize and soybean meal-based diets containing 2958, 3074 and 3163 kcal ME and 22.76, 21.58 and 19.68 percent crude protein, respectively during prestarter (0-14d), starter (15-28d) and finisher (28-42d) phases. We followed the breeder recommendations of vencobb.

• Soya bean used was 46.

• Once agin, Vit pre mix composition was checked. It was 20000000IU only

Nutrient composition calculated or analyzed? Nutrient composition on Calculated values

Chicory root powder composition and its content of inulin and other essential oils or contents if possible. Sir, we have analyzed chicory root powder composition. Now, it has been included in Materials part

BMD composition and concentration and manufacture. Bacitracin Methylene Disalicylate Soluble Powder 50% is a granulated biomass premix containing 150 g (activity) of bacitracin methylene disalicylate per 1 kg of product. Bacitracin Methylene Disalicylate Soluble Powder 50% 

Manufacture - Zoetis

digital pH meter which type its model? Labtronics LT-50 Table-Top Microprocessor Based pH meter

Model No.: LT-50

in table 2 or table 3 use the same groups name Chicory (0.5%) or CRP (0.5%) and not use antibiotic may use BMD.

 Corrected as per reviewer suggestion

if possible insert dates about growth performance table and the effects of these Chicory root powder in feed intake and gain and FCR and other parameters to show effect of these treatments on broiler performance

 Sorry sir, performance and other parameters are going to publish in other journal

Reviewer #2:

Reviewer comments Reply by author

- SBM , add the crude protein level 44 or 46%? Sir, 46 % level

2- MEn, kcal/kg, Ca, digP , not according to the Vencobb requirement. Feed requirements were calculated as per major and minor guides suggestion

Reviewer #3:

Reviewer comments Reply by author

The author stated that in part of material and methods.One bird from each replicate was sacrificed on 42nd day of age from each treatment group. Gut (proventriculus, gizzard, duodenum and ileum) pH was recorded immediately after collection of gastro-intestinal contents from respective part of gut( .

The number of samples of one sample is very, very, very small. We cannot statistically analyze and rely on these results to generalize their results and be confident in their results. The number should not be less than three samples as a minimum.

 Sir, One bird from each replicate means – 8 samples per each treatment.

It is adequate and sufficient for statistical analysis

The figures of jejunum and Ileum were included in manuscript 

The above manuscript has been modified as per reviewer suggestion

---

## [Decision Letter · Decision Letter 1]

1 Nov 2021

PONE-D-21-14465R1SUPPLEMENTATION OF CHICORY ROOT POWDER AS AN ALTERNATIVE TO ANTIBIOTIC GROWTH PROMOTER ON GUT pH, GUT MICROFLORA AND GUT HISTOMORPHOMETERY OF MALE BROILERSPLOS ONE

Dear Dr. srinivas,

Thank you for submitting your manuscript to PLOS ONE. After careful consideration, we feel that it has merit but does not fully meet PLOS ONE’s publication criteria as it currently stands. Therefore, we invite you to submit a revised version of the manuscript that addresses the points raised during the review process.

Please review the referee comments again (about the completion of some methodological shortcomings, the correction of typographical errors, and the clear answer to general questions in terms of intelligibility to the readers) and make your final revision. Thank you

We look forward to receiving your revised manuscript.

Kind regards,

Arda Yildirim, Ph.D.

Academic Editor

PLOS ONE

Journal Requirements:

Additional Editor Comments (if provided):

For your guidance, you can check the reviewers' comments..

Reviewers' comments:

Reviewer's Responses to Questions

**Comments to the Author**

1. If the authors have adequately addressed your comments raised in a previous round of review and you feel that this manuscript is now acceptable for publication, you may indicate that here to bypass the “Comments to the Author” section, enter your conflict of interest statement in the “Confidential to Editor” section, and submit your "Accept" recommendation.

Reviewer #1: All comments have been addressed

Reviewer #2: All comments have been addressed

Reviewer #3: All comments have been addressed

Reviewer #4: All comments have been addressed

Reviewer #5: (No Response)

2. Is the manuscript technically sound, and do the data support the conclusions?

Reviewer #1: Yes

Reviewer #2: Yes

Reviewer #3: Partly

Reviewer #4: Yes

Reviewer #5: Yes

3. Has the statistical analysis been performed appropriately and rigorously? 

Reviewer #1: I Don't Know

Reviewer #2: Yes

Reviewer #3: Yes

Reviewer #4: Yes

Reviewer #5: Yes

4. Have the authors made all data underlying the findings in their manuscript fully available?

Reviewer #1: Yes

Reviewer #2: Yes

Reviewer #3: Yes

Reviewer #4: Yes

Reviewer #5: Yes

5. Is the manuscript presented in an intelligible fashion and written in standard English?

Reviewer #1: Yes

Reviewer #2: Yes

Reviewer #3: Yes

Reviewer #4: Yes

Reviewer #5: Yes

6. Review Comments to the Author

Reviewer #1: in material and methods ad lib from write italic form

Soybean meal type (44-46% cp)- lysine concentration or pure grade

the amount of vitamin A and D in premix is low check the product again.

Coccidiostat which type chemical or ionophores

Chicory root powder proximate analysis (CP, EE, Ash.....) may affect chemical composition of the diet, please to correct total amount to adjust 100% of treated diets

Reviewer #2: Dear Authors Regarding the manuscript title SUPPLEMENTATION OF CHICORY ROOT POWDER AS AN ALTERNATIVE TO ANTIBIOTIC GROWTH PROMOTER ON GUT pH, GUT MICROFLORA AND GUT HISTOMORPHOMETERY OF MALE BROILERS

The scientific background of the topic was well mentioned in the introduction part. The experiment design, as well as the replicates and methods used, were very good. The results obtained were presented in tables well discussed with other author’s results. However, The authors answered all the inquiries and the manuscript may be accepted in this form

Reviewer #3: � He mentioned the following : Record of temperature was maintained on daily basis where the highest daily average temperature recorded is 38.05 °C and the lowest temperature is 18.5 ° C during the experimental period. But the percentage of humidity has not been clarified, and it is known the importance of the relationship between temperature and humidity. Please clarify the degree of humidity during the experiment.

Record of temperature was maintained on daily basis where the highest daily average temperature recorded is 38.05 °C and the lowest temperature is 18.5 ° C during the experimental period. Please move this paragraph from the results and discussion section to the Material and methods.

The data of body weights for the different treatments must be added in the manuscript to clarify the economic return of using supplementation of chicory root powder Is there a clear difference in weight when marketing?

Reviewer #4: ?Why were the growth rates and feed consumption rate for birds not calculated

?Salt means

.What is the percentage of protein in soybean meal? You can be written in the composition table No 1

It is preferable to mention the place of planting the plant, the time of harvest, the method of preservation and drying until analysis.

Reviewer #5: (No Response)

7. PLOS authors have the option to publish the peer review history of their article (what does this mean?). If published, this will include your full peer review and any attached files.

Reviewer #1: **Yes: **Hamada Ahmed

Reviewer #2: No

Reviewer #3: **Yes: **Lamiaa M. Radwan (Poultry Production Department, Faculty of Agriculture, Ain Shams University, Cairo, Egypt )

Reviewer #4: No

Reviewer #5: No

---

## [Author Response · Author response to Decision Letter 1]

3 Nov 2021

Reviewer #1:

Reviewer comments Reply by author

in material and methods ad lib from write italic form

 -------Changed to italic form

Soybean meal type (44-46% cp)- ---- Soyabean meal type has incorporated in Table no 2

lysine concentration or pure grade ----- Sir, Lysine is pure grade 99% and the same was incorporated in Table 2

the amount of vitamin A and D in premix is low check the product again. 

--------Once again, Vit pre-mix composition was checked. Vit A is 20000000IU only and Vit D is 12000 IU

Coccidiostat which type chemical or ionophores ------We have used ‘coxynil’. It is a polyherbal preparation

Chicory root powder proximate analysis (CP, EE, Ash.....) may affect chemical composition of the diet, please to correct total amount to adjust 100% of treated diets ------- Feed requirements were calculated as per major and minor guides suggestion.

Though it was not mentioned, feed given to chicory supplemented groups was adjusted to basal diet requirements. i.e Prestarter (22.76) Starter (21.58) and Finisher (19.68)

Reviewer #2:

Reviewer comments Reply by author

- The scientific background of the topic was well mentioned in the introduction part. The experiment design, as well as the replicates and methods used, were very good. The results obtained were presented in tables well discussed with other author’s results. However, The authors answered all the inquiries and the manuscript may be accepted in this form

 -no comments from reviewer

Reviewer #3

Reviewer comments Reply by author

Record of temperature was maintained on daily basis where the highest daily average temperature recorded is 38.05 °C and the lowest temperature is 18.5 ° C during the experimental period. But the percentage of humidity has not been clarified, and it is known the importance of the relationship between temperature and humidity. Please clarify the degree of humidity during the experiment.

 ----------Included relative humidity

Record of temperature was maintained on daily basis where the highest daily average temperature recorded is 38.05 °C and the lowest temperature is 18.5 ° C during the experimental period. Please move this paragraph from the results and discussion section to the Material and methods. ---------------Moved to material and methods

The data of body weights for the different treatments must be added in the manuscript to clarify the economic return of using supplementation of chicory root powder Is there a clear difference in weight when marketing? Growth performance data has been included in results and discussion part

Reviewer 4

Reviewer comments Reply by author

Why were the growth rates and feed consumption rate for birds not calculated------ Growth performance data has been included in results and discussion part

Salt means ----- Salt means common salt (NaCl) and has included in table 2

What is the percentage of protein in soybean meal? You can be written in the composition table No 1 ------- 46 % CP soyabean meal was used and it has incorporated in feed composition table

It is preferable to mention the place of planting the plant, the time of harvest, the method of preservation and drying until analysis ------Pure Powdered chicory root was directly purchased from Ms Jyothi chicory, Aravind Industries, Guntakal – 515803, Andhra Pradesh, India

The data regarding the cultivation and harvest of Chicory is not available as we have not cultivated.

Reviewer 5

Reviewer comments Reply by author

• However, it is better if the author includes growth performance of the birds. The number of birds used in the experiment was also insufficient. -------- Growth performance data has been included in results and discussion part

• The author stated that one bird per replica was sacrificed which is very small and statistically to rely on. The minimum sample size for statistical analysis should be three. -------Sir, One bird from each replicate means – 8 samples per each treatment.

It is adequate and sufficient for statistical analysis (8 samples per treatment)

• Why the author focussed only on male broiler birds? Why you did not include female broiler birds? -------- male birds were selected to get uniform growth rate among the groups and to reduce error variation.

Growth performance data was added in results section and two references were newly added in reference section.

1. Ortiz LT, Rodri guez ML, Alzueta C, Rebole A, Trevin J. Effect of inulin on growth performance, intestinal tract sizes, mineral retention and tibial bone mineralization in broiler chickens. British Poultry Science 2009; 50 (3): 325-332.

2. Yousfi Z, Kazemi Fard M, Rezaei M, Ansari P Z. Effect of chicory extract and probiotic on performance, caracas characteristics, blood parameters, intestinal microflora and immune response of broiler chickens. 2017; (9): 185-195.

The above manuscript has been modified as per reviewer suggestion

---

## [Decision Letter · Decision Letter 2]

22 Nov 2021

SUPPLEMENTATION OF CHICORY ROOT POWDER AS AN ALTERNATIVE TO ANTIBIOTIC GROWTH PROMOTER ON GUT pH, GUT MICROFLORA AND GUT HISTOMORPHOMETERY OF MALE BROILERS

PONE-D-21-14465R2

Dear Dr. srinivas,

We’re pleased to inform you that your manuscript has been judged scientifically suitable for publication and will be formally accepted for publication once it meets all outstanding technical requirements.

Kind regards,

Arda Yildirim, Ph.D.

Academic Editor

PLOS ONE

Additional Editor Comments (optional):

Reviewers' comments:

Reviewer's Responses to Questions

**Comments to the Author**

1. If the authors have adequately addressed your comments raised in a previous round of review and you feel that this manuscript is now acceptable for publication, you may indicate that here to bypass the “Comments to the Author” section, enter your conflict of interest statement in the “Confidential to Editor” section, and submit your "Accept" recommendation.

Reviewer #2: All comments have been addressed

Reviewer #3: All comments have been addressed

Reviewer #4: All comments have been addressed

Reviewer #5: All comments have been addressed

2. Is the manuscript technically sound, and do the data support the conclusions?

Reviewer #2: Yes

Reviewer #3: Yes

Reviewer #4: Partly

Reviewer #5: Yes

3. Has the statistical analysis been performed appropriately and rigorously? 

Reviewer #2: Yes

Reviewer #3: Yes

Reviewer #4: I Don't Know

Reviewer #5: (No Response)

4. Have the authors made all data underlying the findings in their manuscript fully available?

Reviewer #2: Yes

Reviewer #3: Yes

Reviewer #4: Yes

Reviewer #5: (No Response)

5. Is the manuscript presented in an intelligible fashion and written in standard English?

Reviewer #2: Yes

Reviewer #3: Yes

Reviewer #4: Yes

Reviewer #5: (No Response)

6. Review Comments to the Author

Reviewer #2: The scientific background of the topic was well mentioned in the introduction part. The experiment design, as well as the replicates and methods used, were very good. The results obtained were presented in tables well discussed with other author’s results. However, The authors answered all the inquiries and the manuscript may be accepted in this form

Reviewer #3: (No Response)

Reviewer #4: Why was no analysis done of the bioactive components of the additive used in the experiment?

Why was the humidity level not made clear during the experiment? Including relative humidity

Why was an economic assessment such as the European production efficiency factor not done?

Reviewer #5: (No Response)

7. PLOS authors have the option to publish the peer review history of their article (what does this mean?). If published, this will include your full peer review and any attached files.

Reviewer #2: **Yes: **Prof. Ahmed Ali Saleh

Reviewer #3: **Yes: **Lamiaa M. Radwan (Poultry Production Department, Faculty of Agriculture, Ain Shams Univ., Cairo, Egypt )

Reviewer #4: **Yes: **Gamal Ali Abdelhafez Ali Hamady

Reviewer #5: No

---

## [Editor Report · Acceptance letter]

1 Dec 2021

PONE-D-21-14465R2 

Supplementation of chicory root powder as an alternative to antibiotic growth promoter on gut pH, gut microflora and gut histomorphometery of male broilers 

Dear Dr. Gurram:

I'm pleased to inform you that your manuscript has been deemed suitable for publication in PLOS ONE. Congratulations! Your manuscript is now with our production department. 

Kind regards, 

on behalf of

Prof. Dr. Arda Yildirim 

Academic Editor

PLOS ONE